# Influence of Varying Amounts of Alumina (Al$_2$O$_3$) on the Wear Behavior of ZnO, SiO$_2$ and TiO$_2$ Compounds

**Ali Ihsan Kaya** [ORCID]

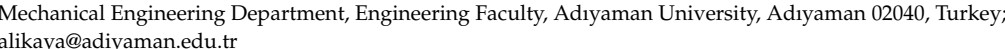

Mechanical Engineering Department, Engineering Faculty, Adıyaman University, Adıyaman 02040, Turkey; alikaya@adiyaman.edu.tr

**Abstract:** This study aimed to exploit the superior properties of TiO$_2$, ZnO, SiO$_2$ and Al$_2$O$_3$ inorganic materials to combine them under pressure and investigate their mechanical properties. The hot pressing technique was used to produce new materials. Varying amounts of alumina such as 0, 5, 10, 20 and 30 wt% in compounds was considered. The produced materials were characterized by SEM, EDS and XRD analyses. The microhardness properties of the materials were studied, and their tribological properties under different wear loads, i.e., 10 N, 20 N and 30 N, were investigated for every specimen. In XRD analysis, it was observed that no significant new peaks were formed regarding increasing alumina content. The SEM and EDS characterization analyses showed that the materials had a two-phase structure with complex boundaries, and no clear grain boundaries were formed. Moreover, the elements in the EDS analyses and the compounds in the XRD analyses were found to be in line with each other. In wear tests, it was seen that as the wear load increased, the depth and width of the wear track increased. The highest weight loss under different wear loads was obtained for the Ti55Si15Zn20Al10 material. It was determined that as the Al$_2$O$_3$ wt% increased over 10 wt%, the weight losses decreased. It was observed that there was an increase in the microhardness value generally depending on the increase of alumina content in compounds.

**Keywords:** Al$_2$O$_3$; sol–gel; wear; powder metallurgy; hot pressing

## 1. Introduction

Powder metallurgy is a widely used production technique to obtain ready-to-use predetermined shapes or properties of an engineering component without further processing. The characteristics of a produced material are highly dependent on the starting powders. Not only metallic but also nonmetallic mixed constituents can be used in the finished product. TiO$_2$, ZnO, SiO$_2$ and Al$_2$O$_3$ materials are commonly used in this method.

Particle size is one of the important factors in powder metallurgy. For example, the influence of different particle sizes of titanium dioxide (TiO$_2$) on the reinforcement of the Al-TiO$_2$-C system was studied [1]. It was found that the particle size promotes sintering, changes the reaction mode and affects the distribution of reinforcement in Al-TiO$_2$-C system. Ghasali et al. [2] found that the microwave method was the best one among other production methods because of the formation of Al$_3$Ti phases at low sintering times in terms of microstructure and mechanical properties. Islak et al. [3] reported that when increasing the TiO$_2$ powder, pore contents were decreased in Al$_2$O$_3$-TiO$_2$ composite coatings and microhardness values were improved compared with the substrate material. Kumar et al. [4] reported that the Al6061 metal matrix with a 3 wt% TiO$_2$-reinforced composite showed superior physical, mechanical and resistance to wear properties. Joshua et al. [5] surveyed the effect of nano-TiO$_2$ particles on the AA7068 metal matrix composites and an increase in microhardness was found depending on TiO$_2$ increase. Nayak et al. [6] studied the synergistic effect of Al$_2$O$_3$ and TiO$_2$ nano-filled in glass fiber-reinforced composites and found that these nano-fillers improved the interlaminar shear strength and strain but reduced the storage and loss modulus of glass fiber-reinforced polymer composites.

Sathiyakumar and Gnanam [7] found that increasing the amount of $TiO_2$ up to the solubility limit in alumina increases the flexural strength of materials, but a sudden decrease occurred after the solubility limit.

In biomedical applications, Ti and its alloys are commonly utilized as medical implants for a variety of purposes, whereas ZnO material is used in pharmaceutical and cosmetic uses. Sadooghi and Hashemi [8] found that the density of CuO, ZnO and $Al_2O_3$ nanoparticles added to Al matrix composites were greater; the lowest hardness was in the case of ZnO-doped composites, the maximum wear rate was with pure Al matrix and flexural strength was improved with the addition of mentioned nanoparticles. Kamitakahara et al. [9] stated that the chemical durability of glass ceramics was increased, but apatite-forming ability was decreased with the increase of ZnO. Nguyen et al. [10] surveyed the effect of $SiO_2$, $TiO_2$, $Fe_2O_3$, ZnO and clay nanoparticles in epoxy coatings. They stated that nanoparticles tend to occupy the voids and nano-ZnO enhanced the thermal stability more than others, and $TiO_2$ nanoparticles revealed the best improvement in terms of impact strength. Rubio-Marcos et al. [11] observed a decrease in the piezoelectric coefficient and a linear reduction of the dielectric constant and an increase of the ferroparaelectric phase transition temperature of piezoceramic material regarding ZnO doping. Yuan et al. [12] revealed that 46% and 47% enhancement were obtained in the fracture strength and toughness of the PZT piezoelectric composites in the case of 2 wt% doped ZnO whiskers compared with monolithic PZT ceramics. It was found that ZnO nanoparticles enhanced the tribological behavior of ionic liquids better than CuO nanoparticles [13].

Because of their biocompatibility and beneficial biological effects, silica-based materials are another biomaterial widely used in biomedical applications such as $TiO_2$ and ZnO. Spherical-shaped silica ($SiO_2$) nanoparticle-doped polymer composites exhibit high mechanical properties and corrosion resistance [14–17]. Hussain et al. [18] found that the structural, morphological, magnetic and electrical properties of Sr-hexa ferrite systems are strongly affected by incorporating $SiO_2$ particles. Deka et al. [19] stated that with the addition of $SiO_2$, ZnO and nano-clay nanoparticles to wood plastic composites, the mechanical, hardness and thermal properties were improved and the water absorption capacity of composites was reduced. Larsen et al. [20] investigated the addition of silica to $MgO$-$Al_2O_3$-$P_2O_5$ glass to examine the chemical durability of glasses and stated that in order to improve the stability of glasses significantly, 30 mol% addition of $SiO_2$ was required, but that amount would lead to a low thermal expansion coefficient. Zheng and Wang [21] studied the mechanical properties of $SiO_2$-doped boron nitride coatings, and the results indicated that with $SiO_2$ powder inclusion, twice as much strength, attributed to the more active surface of that nano-$SiO_2$, was acquired than with pure boron nitride fibers.

Unlike $TiO_2$, ZnO and $SiO_2$, alumina was the first biomaterial to be used in clinical applications because of its excellent biocompatibility, hardness, strength to withstand fatigue and corrosion resistance. Agrawal and Satapathy [22] found a substantial improvement in effective thermal conductivity and glass transition temperature but a decrease in the coefficient of thermal expansion and void content in epoxy and polypropylene with the increase of $Al_2O_3$ filler. Dhara et al. [23] found that a higher glass transition temperature of barium borosilicate than with sodium borosilicate glass was obtained with $Al_2O_3$ doping. This phenomenon was attributed to more connected $BO^{4-}$ structural units. Santos et al. [24] stated that a linear hardness increase, as opposed to fracture toughness, was obtained with the addition of alumina to $ZrO_2$-$Al_2O_3$ ceramic composites. Zhou et al. [25] reported that $Al_2O_3$ addition was very effective in promoting the mechanical strength and thermal expansion coefficient of $Ta_2O_5$ ceramics. Farvizi et al. [26] observed that the wear resistance of the $Al_2O_3$-doped nanocomposites was much better than pure NiTi and a remarkable increase in the hardness and elastic modulus was obtained. Spencer et al. [27] reported that the metallurgical bonding and hardness of 316 austenitic stainless steel were improved with the addition of $Al_2O_3$, but the corresponding wear rate was decreased. Bazrgari et al. [28] reported that 1% alumina incorporation into epoxy was optimum for impact strength, stiffness, flexural strength and wear resistance. Khalil et al. [29] stated that an

improvement in wetting behavior was obtained with the inclusion of $Al_2O_3$ nanoparticles in the epoxy matrix and a significant improvement of both storage and loss modulus was achieved with 0.5 wt% alumina. The synergistic effect of $Al_2O_3$ and $B_4C$ on the Al 5059 was investigated [30]. It was found in a study that $Al_2O_3$-reinforced composites showed better wear properties than Al–12Si–CuNiMg matrix alloy [31]. Besides, the effect of $Al_2O_3$ particles on the properties of the Al matrix structures was extensively discussed in a review study [32].

It could be stated that a high refractive index, low cost, good optical transparency, non-toxicity and chemical stability constitute the superior aspects of the $TiO_2$ compound. In addition, compared with carbon-based nano-fillers, $TiO_2$ and $Al_2O_3$ have superb low density, weathering properties, thermal and mechanical properties and low manufacturing cost [6]. Besides, ZnO has superior biological, chemical and optical properties that could be used in UV radiation [33]. The thermal resistance, small size, strong surface energy and relative inertness of silica are the reasons behind its utilization in different applications [34]. Furthermore, low thermal conductivity, excellent electrical insulation, low cost and good chemical stability properties could be expressed for alumina [35]. Because of their distinguished properties, these powders were the subjects of research in this study to benefit from the compounds.

When examining the studies on biomaterials, it is evident that $TiO_2$, ZnO, $SiO_2$ and alumina make considerable contributions to this field in terms of both chemical structure and mechanical properties. However, to the best of our knowledge, no academic study has been reported in which the properties of these four biomaterials have been investigated by combining them together. In order to contribute to the literature, in this study, different amounts of $TiO_2$, ZnO, $SiO_2$ and $Al_2O_3$ materials, widely used in different applications and having great promise in many fields, were used to produce new materials by powder metallurgy to provide better performance because of their synergistic effects [36]. The effect of varying amounts of alumina such as 0, 5, 10, 20, and 30 weight percent (wt%) in compounds was examined. The produced materials were characterized by x-ray diffractometry (XRD), scanning electron microscopy (SEM) and energy-dispersive spectroscopy (EDS) analyses. The microhardness properties of the materials were also studied, and tribological properties of the materials under different wear loads, i.e., 10 N, 20 N and 30 N, were investigated for every specimen. It was seen that the weight loss generally decreased in the wear test depending on the increase in $Al_2O_3$ content in $TiO_2$, $SiO_2$ and ZnO compounds. Besides, it was found that the wear track and depth were proportional to the wear load increase.

## 2. Materials and Methods

### 2.1. Production of Materials

The materials, whose weight percentages are given in Table 1, were produced by the hot pressing method. $TiO_2$ (99% purity, AppliChem, Monza, MB, Italy), $SiO_2$ (99% purity, Sigma Aldrich), ZnO (99% purity, Fluka, Buchs, Switzerland) and $Al_2O_3$ (99% purity, Sigma Aldrich, St. Louis, MO, USA) compounds were utilized during production. Different temperature and pressure values were used until an exemplary structure of these materials was obtained. All the materials with diameters of 2 mm and heights of 2 mm were unidirectionally hot pressed under a force of 45 MPa and a temperature of 950 °C. A vacuum pressure of $10^{-4}$ mbar was applied to the materials during production. Then, as shown in Figure 1, the temperature was increased to 950 °C over 10 min; the materials were left to rest under 45 MPa at 950 °C for 30 min before being cooled to room temperature over 5 min. Five materials were considered according to varying amounts of alumina, and the wt% of every compound in composition is given in Table 1.

**Table 1.** wt% of materials used in production.

| Materials | TiO$_2$ (wt%) | SiO$_2$ (wt%) | ZnO (wt%) | Al$_2$O$_3$ (wt%) |
|---|---|---|---|---|
| Ti60Si15Zn20Al5 | 60 | 15 | 20 | 5 |
| Ti55Si15Zn20Al10 | 55 | 15 | 20 | 10 |
| Ti45Si15Zn20Al20 | 45 | 15 | 20 | 20 |
| Ti35Si15Zn20Al30 | 35 | 15 | 20 | 30 |
| Ti60Si20Zn20 | 60 | 20 | 20 | - |

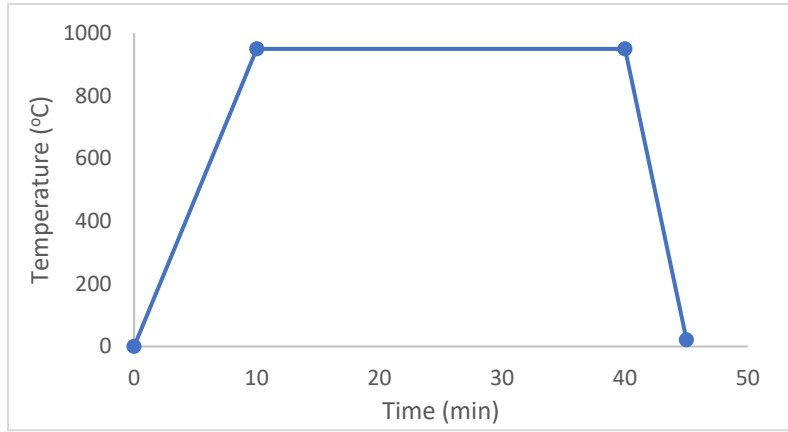

**Figure 1.** Hot pressing production phases.

As can be seen in Table 1, new materials were produced by considering the weight percentages of different biomaterials in this study. Therefore, depending on this calculation, one of the existing materials must be reduced by weight in order to look at the effect of another doped material. Since the addition of ZnO improves wear resistance and silica improves mechanical properties, we tried to keep the ratios of these materials constant as possible. For this reason, one material, in this case TiO$_2$, was considered as a matrix structure and its ratios were changed as per alumina addition. In addition, while producing biomaterials, care was taken to ensure that the weight percentages of the additives were lower by weight than the TiO$_2$ determined as the matrix.

### 2.2. Characterization

XRD, SEM and EDS analysis devices were used to analyze the microstructures and chemical properties of the materials. Rigaku Miniflex600 was used for x-ray diffractometry analysis. Samples were irradiated with Cu-K$\alpha$ radiation at 40 kV voltage and 15 mA current to obtain the diffraction patterns of materials. Moreover, XRD analyses were conducted at a wavelength of 1.5406 ($\lambda$) between 10° and 90° with a step speed of 0.02° and a rate scanning speed of 2° per minute. JEOL JSM 7001F and Oxford INCA devices were utilized for SEM and EDS analyses, respectively.

### 2.3. Pin-on-Disc Wear Testing

All materials were subjected to a pin-on-disc wear test. A tribometer T10/20 device operating according to ASTM G99-17 was used for the relevant test. The materials were 2 mm in diameter and 2 mm in height. A 6 mm diameter stainless steel ball was used for the pin-on disc wear test. The materials were weighed before and after each wear test with weighing equipment having an accuracy of $10^{-4}$ g. After every experiment, the ball's friction point was changed in order to have a non-abraded contact point on the specimen. Weight losses of all materials under 10 N, 20 N and 30 N wear loads were determined in wear tests. The sliding distance, rotational speed and sliding speed were set to 300 m, 240 rpm and 0.005 m/s, respectively.

*2.4. Microhardness Measurement*

After the wear tests, Vickers hardness values were determined by applying a constant force to the back surfaces of the materials by using Qness Q10M equipment. The surfaces of the materials were abraded with the help of abrasive papers because of the highly variable topography. Therefore, the surfaces of all materials were treated with 100, 400, 600, 800, 1000 and 1200 silicon carbide abrasive papers, respectively. The surfaces of the materials were polished with appropriate polishing paper. Hardness values were obtained by applying a 3000 g (HV3) load under 10 s dwelling time, and the microhardness of each material was repeated five times. Each measurement site was chosen at a sufficient distance from the previous points so that the relevant measurement would not be affected by the other measurements. The average of the five hardness values was considered for evaluation and comparison.

## 3. Results

*3.1. XRD*

X-ray diffraction analysis is a form of quantitative analysis used for multi-phase materials [37]. It is a non-destructive testing technique allowing for characterizing the metals, minerals, etc. in terms of the position of atoms, their arrangement in each unit cell and spacing between the atomic planes [38]. The peak position and intensity identification of the unique crystal structure of a material can be determined through X-ray diffraction. Therefore, the crystallinities, phases and peaks of hot-pressed materials were examined by the X-ray diffraction technique. XRD analyses were conducted between 10° and 90°, and the results are shown in Figure 2. In order to be able to compare diffraction patterns of all materials in the same graph, all the XRD intensity values of each material were normalized according to the biggest intensity (cps deg.) value, and the results were plotted as per the normalized value. In order to see the peak changes that occur depending on the alumina content, the intensity of peak values of each material is also shown on the graph. Crystal structure analysis results of compounds were compared with standard card numbers to determine what phases were formed. As seen in Figure 2, the peak values of each compound matched to Crystallography Open Database (COD) standard card numbers indicated on the graph by using different geometrical shapes to compare the results.

In Figure 2, it was seen that the formed phases (rutile, quartz, zincite, etc.) contained the starting elements. When the XRD graph of the Ti60Si20Zn20 material was examined, it was observed that the most severe peaks occurred between 2θ~26.44° and 26.82°. Besides, it was found that as the alumina content increased by weight, the intensity of the peak at 2θ~26.44° decreased in general, and the peak intensity of 2θ~26.82° maintained its intensity compared with 2θ~26.44°. The sharp peaks in all XRD patterns indicate the crystalline structure positions of different phases in the materials [38]. Therefore, it can be emphasized that the compounds in 2θ positions have more a crystalline structure than other phases because of the sharpness of the peak intensities. In addition, it cannot be stated that there was a significant increase or decrease in the intensity of the peaks, except for 2θ~26.44° in the XRD graphs depending on alumina content. The intensity peak values at 2θ~26.44°–26.82° were closest to each other in the Ti55Si15Zn20Al10 material. Nevertheless, it could be stated from Figure 2 that there no significant new peaks occurred because of the increase in alumina content, and the materials showed similar peaks.

In Table 2, the lattice parameters and standard DB card numbers of formed phases are given. XRD results showed that rutile, spinel, quartz, sillimanite and kyanite phases were found in the first material. The second material's XRD results showed that quartz, mullite and spinel phases were encountered. The third material's XRD results showed that spinel, zincite, rutile, cristobalite and quartz phases were present. The fourth material phases were rutile, gahnite, cristobalite and zincite. The fifth material's XRD results showed that rutile, gahnite, quartz and cristobalite phases were encountered. Some of the related phases were found in a study of a $SiO_2$-$Al_2O_3$-MgO/ZnO glass ceramic system investigating its sinterability and

mechanical properties [39]. To name a few other papers, refs. [40,41] could be mentioned as containing some of the related phases which were encountered in this study.

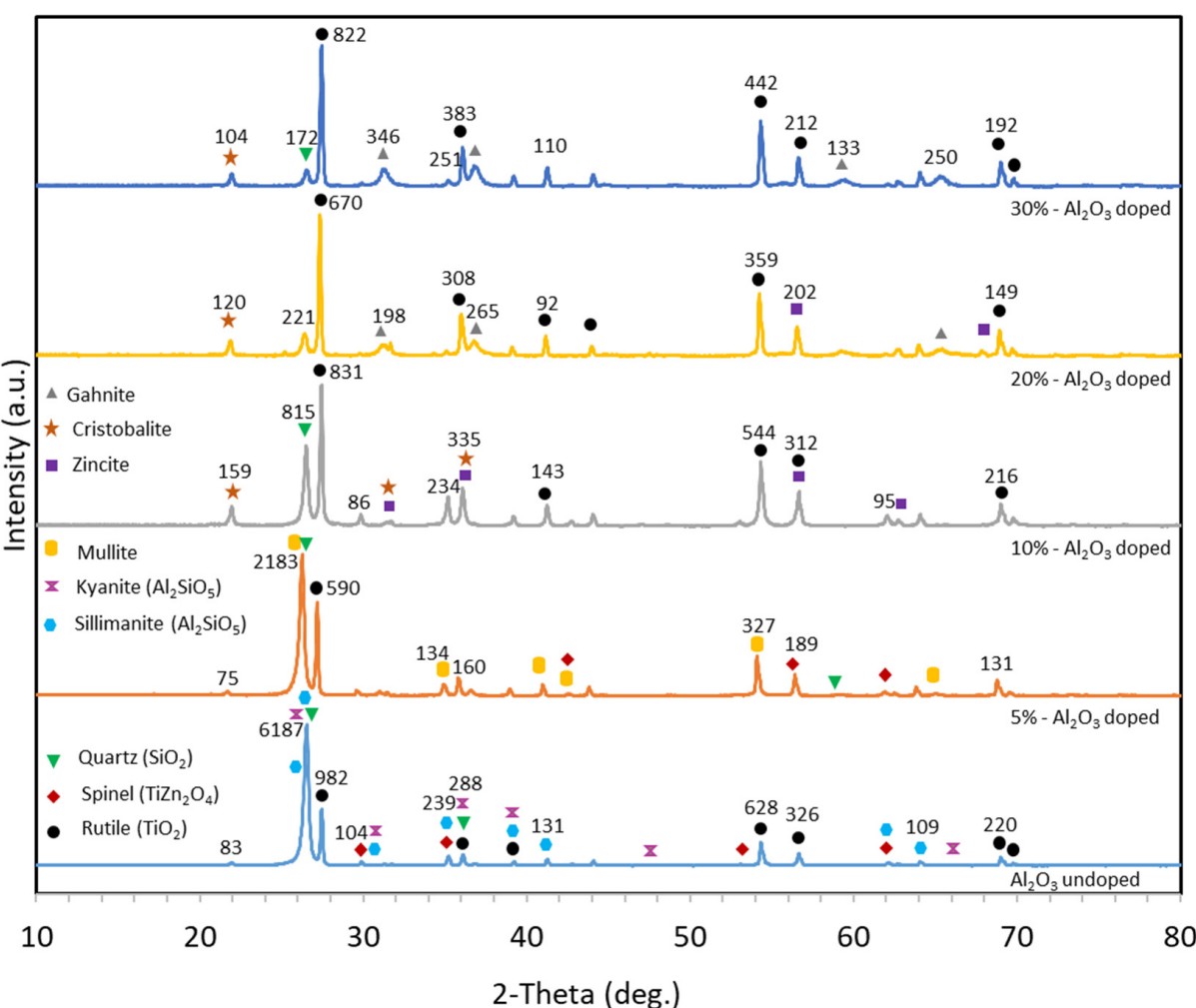

**Figure 2.** XRD results of materials containing varying amounts of $Al_2O_3$ (a.u. stands for arbitrary units).

**Table 2.** Lattice parameters and DB card numbers of the phases.

| Card Number | Compound | Lattice Parameters | | |
|---|---|---|---|---|
| | | a (Å) | b (Å) | c (Å) |
| 9015662 | Rutile ($TiO_2$) | 4.599923 | 4.599923 | 2.963614 |
| 9001693 | Spinel ($TiZn_2O_4$) | 8.458678 | 8.458678 | 8.458678 |
| 9012600 | Quartz ($SiO_2$) | 4.956337 | 4.956337 | 5.433133 |
| 9000714 | Sillimanite ($Al_2SiO_5$) | 7.475310 | 7.713990 | 5.700429 |
| 9001835 | Kyanite ($Al_2SiO_5$) | 7.052520 | 7.772450 | 5.515804 |
| 7105575 | Mullite ($Al_6Si_2O_{13}$) | 7.564156 | 7.687715 | 2.886860 |
| 9008877 | Zincite (ZnO) | 3.255732 | 3.255732 | 5.221811 |
| 9001578 | Cristobalite ($SiO_2$) | 5.002741 | 5.002741 | 6.908230 |
| 9013642 | Gahnite ($Al_2ZnO_4$) | 8.084869 | 8.084869 | 8.084869 |

### 3.2. SEM-EDS Analysis

SEM images and EDS results of Ti60Si15Zn20Al5 are given in Figure 3a,b, respectively. As shown in Figure 3a, it was observed that two phases with different colored structures were formed. It could be stated that the secondary phase and the matrix phase constitute approximately half of the whole structure but show a heterogeneous distribution in Figure 3a. Therefore, it was observed that a layered structure of two different phases

could be seen from the different colors that constitute the structure. There was no clear grain structure between these phases and complex boundaries were formed [42,43]. To comprehensively analyze the elements in the Ti60Si15Zn20Al5 material, a large yellow rectangle area in Figure 3a was taken from the middle region and general EDS analysis was performed. The EDS analysis results in Figure 3b revealed that Ti, O, Zn, Si and Al elements were present for the Ti60Si15Zn20Al5 material. Among these materials, Ti (51%) and O (37%) elements were dominant in proportion.

SEM images and EDS results of Ti55Si15Zn20Al10 are given in Figure 3c,d, respectively. As clearly seen from the different colors, two different structures occurred, as shown in Figure 3a [42]. To analyze the elements of the Ti55Si15Zn20Al10 specimen, a large yellow rectangle in Figure 3c was taken from the middle region and general EDS analysis was performed. According to the EDS analysis results in Figure 3d, it was found that the Ti55Si15Zn20Al10 material contained Ti, O, Zn, Si and Al elements and Ti (48%) and O (44%) elements were dominant.

SEM images and EDS results of Ti45Si15Zn20Al20 material are given in Figure 3e,f, respectively. A large yellow rectangle in Figure 3e was chosen from the middle region and general EDS analysis was performed. It was found that the Ti45Si15Zn20Al20 material contains Ti, O, Zn, Si and Al elements (Figure 3f) and Ti (48 %) and O (44%) elements were dominant, and the Al ratio was higher than Ti60Si15Zn20Al5 and Ti55Si15Zn20Al10 materials. In addition, two small points (yellow asterisks in Figure 3e) were chosen not only from the white region but also from the black region for EDS analysis. It was found that the ratios of Ti, O and Zn elements were approximately 48%, 47% and 2% in the white region and 76%, 1% and 22% in the black region, respectively.

SEM image and EDS analysis results of the alumina-free specimen are given in Figure 3g,h, respectively. SEM images of the Ti60Si20Zn20 material in Figure 3g are similar to the results of the Ti60Si15Zn20Al5 material in Figure 3a. To analyze the elements of the Ti60Si20Zn20 material, the yellow rectangle shown in Figure 3g was taken to perform a general EDS analysis. As seen from Figure 3g, it was found that the Ti60Si20Zn20 material contains Ti, O, Zn and Si elements but not Al element as expected. According to the EDS analysis results, Ti (52%) and O (37%) elements were dominant in proportion.

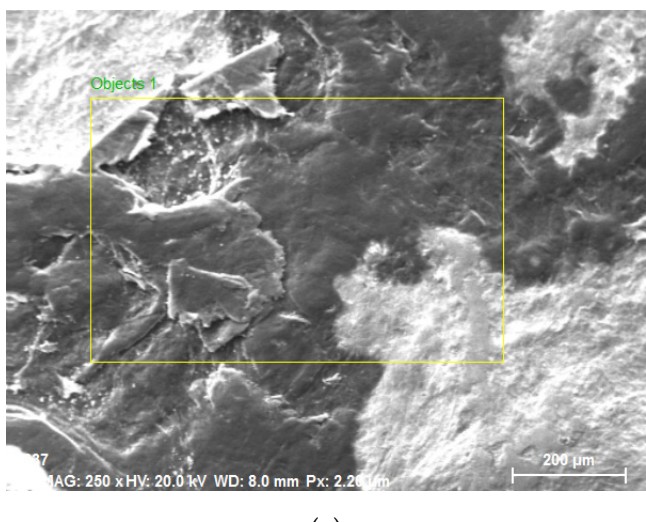

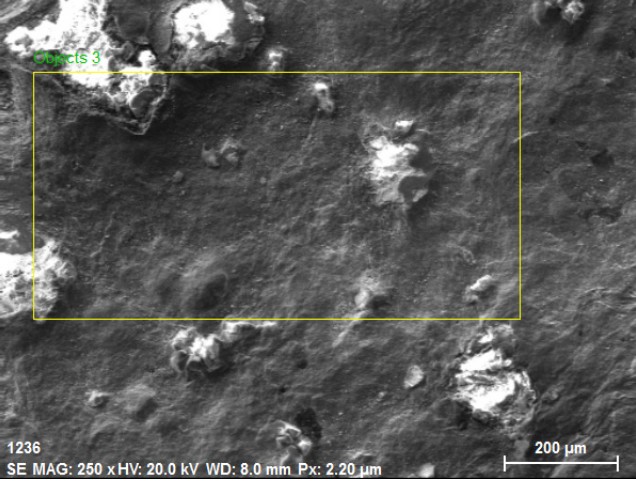

(a)                                                        (c)

**Figure 3.** *Cont.*

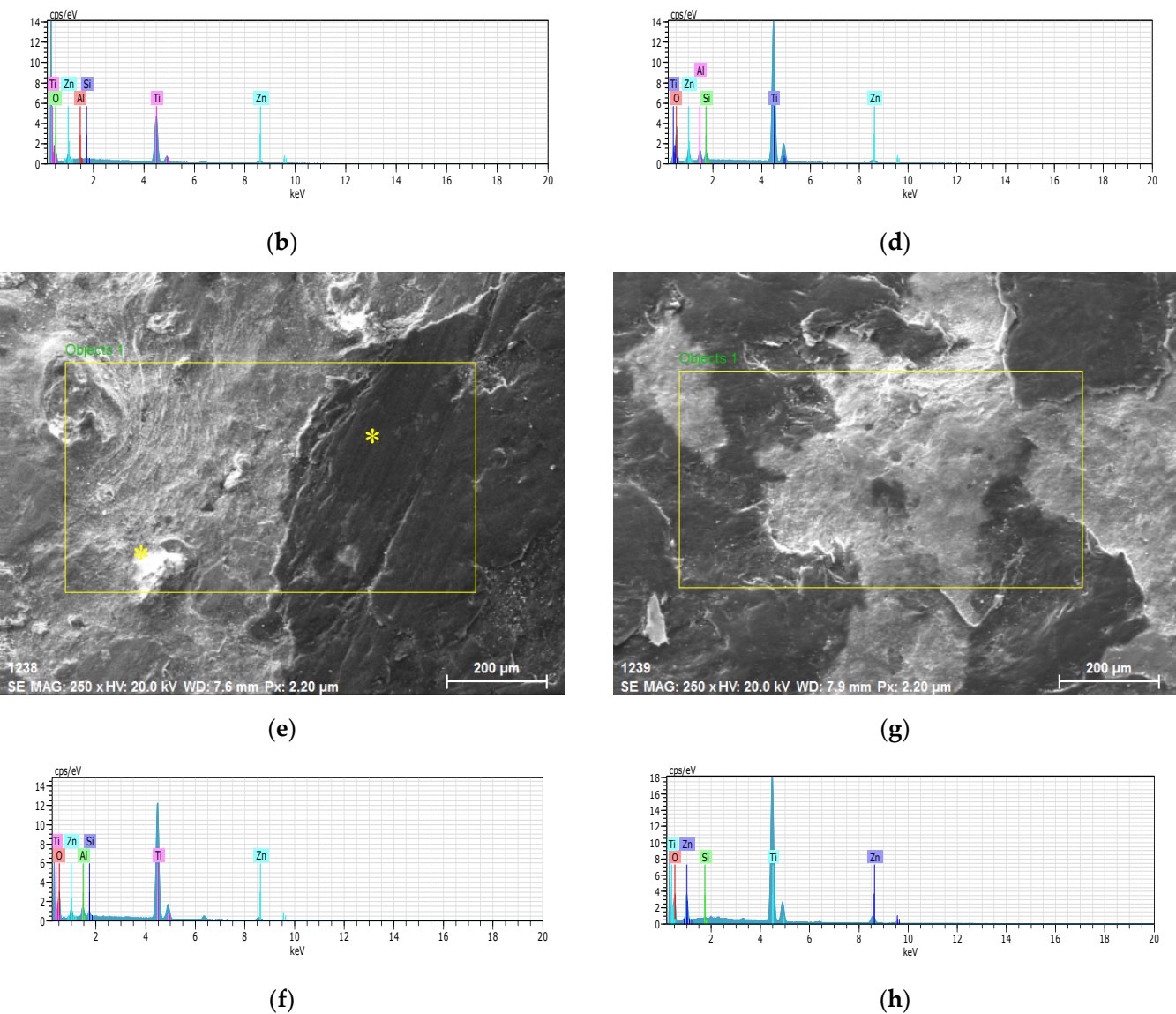

**Figure 3.** Ti60Si15Zn20Al5 material's (**a**) SEM image (**b**) EDS analysis, Ti55Si15Zn20Al10 material's (**c**) SEM image and (**d**) EDS analysis, Ti45Si15Zn20Al20 material's (**e**) SEM image and (**f**) EDS analysis and Ti60Si20Zn20 material's (**g**) SEM image and (**h**) EDS analysis results.

*3.3. Wear Test and Weight Losses*

In this study, varying amounts of alumina effect in $TiO_2$, $ZnO$, $SiO_2$ compounds were considered to investigate the wear characteristics of the hot-pressed materials under 10 N, 20 N and 30 N wear loads. The sliding distance, rotational speed and sliding speed were set to 300 m, 240 rpm and 0.005 m/s, respectively. All materials were weighed before being subjected to the wear test, and the weight losses of the materials were measured with appropriate equipment having a precision of $10^{-4}$ g after each wear test. The wear resistance of each material was interpreted based on the weight losses in question. The worn surfaces of materials were examined by using optical microscopy, SEM, EDS and XRD equipment.

SEM images of the Ti60Si15Zn20Al5 material under 10 N, 20 N and 30 N wear loads are given in Figure 4a. At 10 N wear load, it was observed that there was flattening from region to region due to friction, and the wear tracks were barely visible. Besides, it was determined that the wear track width of the Ti60Si15Zn20Al5 material increased, and plowed structures (point "a" in Figure 4a) were formed due to the rupture occurrences in the case of 20 N wear load. At 30 N wear load, it was further observed that the highest

wear track width occurred. The amount and size of the flattened wear surfaces were due to friction and the plowed structures formed because of ruptures (point "b" on Figure 4a) increased compared with other loads.

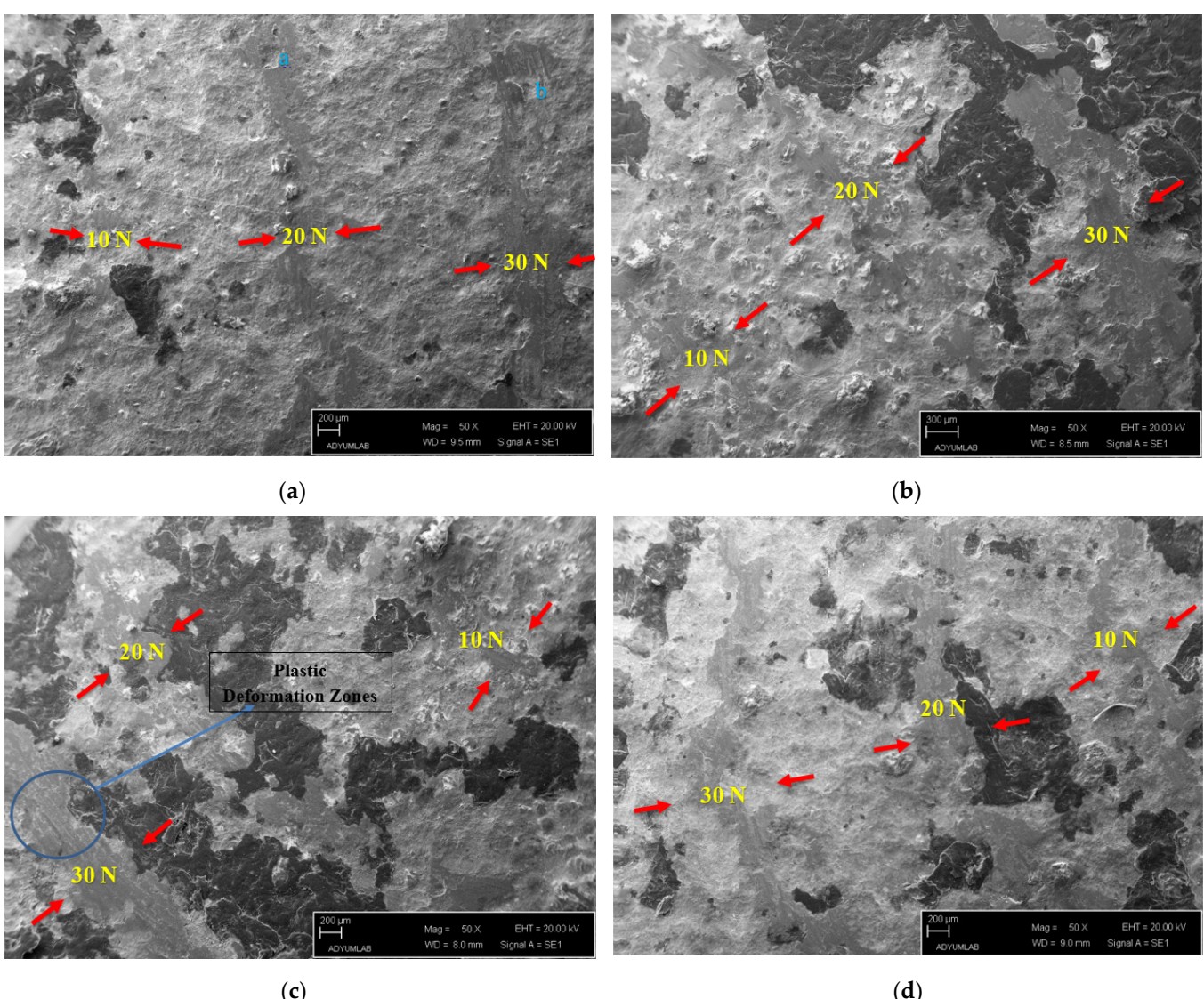

**Figure 4.** (**a**) SEM of worn Ti60Si15Zn20Al5 material, (**b**) SEM of worn Ti55Si15Zn20Al10 material, (**c**) SEM of worn Ti45Si15Zn20Al20 material and (**d**) SEM of worn Ti60Si20Zn20 material.

In Figure 4b, a post-wear SEM image of the Ti55Si15Zn20Al10 material under 10 N, 20 N and 30 N wear loads is given. It was observed that the wear track under 10 N wear load was barely visible, as in Figure 4a, but more apparent. Besides, the wear track depth was lesser than the Ti60Si15Zn20Al5 material at the same wear load. At 20 N wear load, it was observed that the wear track depth was increased compared with the 10 N wear load for the Ti55Si15Zn20Al10 material; however, the wear track width was smaller than the Ti60Si15Zn20Al5 material. It was further observed in Figure 4b that the highest wear track depth and width were obtained at 30 N wear load. Moreover, the amount and size of flattening surfaces increased significantly because of higher friction load compared with 10 N and 20 N wear loads.

In Figure 4c, a post-wear SEM image of the Ti45Si15Zn20Al20 material is given. It was observed that very small flattened structures occurred because of friction, and the wear zone was hardly visible because of the small depth of the wear track under 10 N wear load. The wear track and its depth were more evident at 20 N wear load than under

10 N wear load. It could also be stated that fewer plowed areas were formed than with the Ti60Si15Zn20Al5 and Ti55Si15Zn20Al10 materials. At 30 N wear load, the depth and width of the wear track were more significant than both 10 N and 20 N wear loads for the Ti45Si15Zn20Al20 material. In addition, wavy, grooved structures of plastic deformation zones were formed in the wear track region because of the abundance of the parts broken off owing to the friction remaining in the wear track trajectory at 30 N wear load compared with 10 N and 20 N wear loads [44,45]. Moreover, as in other materials, it was observed that as the wear load was increased, the wear track and depth were increased.

In Figure 4d, a post-wear SEM image of alumina-free Ti60Si20Zn20 material is given. The friction-related flattening regions of the Ti60Si20Zn20 material under 10 N wear load were observed to be higher than the previously given materials. It could be stated that under the 20 N wear load, the depth and width of the wear track were greater than under the 10 N wear load. Moreover, under the 30 N wear load, wear track depth and width were higher than those in the other materials. It was observed that as the wear load was increased, the wear track and depth were increased, as in the other materials. In addition, because of the dominance of secondary phase regions on the surface of Ti60Si20Zn20 material, it was observed that the number of plowed structures in the wear trajectory was higher than other materials under every wear load because of the ruptures from the secondary phase regions.

Weight losses of hot-pressed materials were measured after every wear test under different wear loads. The mentioned weight losses after the wear tests are given in Figure 5. When the graph was examined, it was observed that the lowest weight loss under 10 N wear load was realized in the Ti60Si20Zn20 material. On the contrary, the highest weight loss occurred in the Ti55Si15Zn20Al10 material under the same wear load. At 20 N wear load, it was determined that the lowest weight losses were observed in Ti60Si15Zn20Al5 and Ti35Si15Zn20Al30 materials, respectively whereas the highest weight loss occurred in the Ti55Si15Zn20Al10 material. At 30 N wear load, it was determined that the lowest weight losses were observed in Ti60Si15Zn20Al5 and Ti35Si15Zn20Al30 materials, respectively, whereas the highest weight loss was observed in Ti55Si15Zn20Al10 material, as in other wear loads. When each material in Figure 5 was examined in terms of weight losses under different wear loads, it could be stated that the highest and lowest weight losses generally occurred under 30 N and 10 N wear loads, respectively. In addition, when it was further examined, it was determined that there was generally a linear relationship between wear loads and weight losses, except for the Ti60Si15Zn20Al5 material.

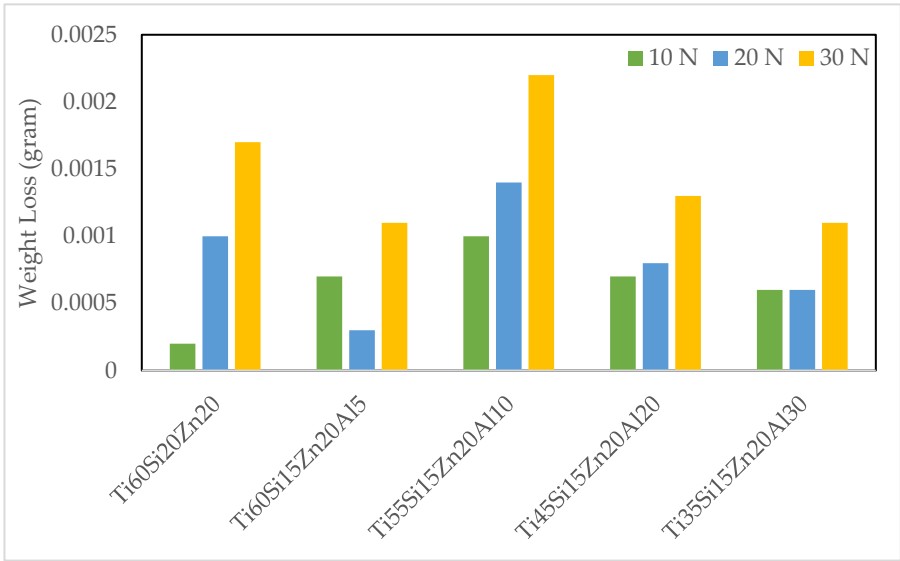

**Figure 5.** Weight losses of specimens according to wear loads.

### 3.4. Friction Coefficients

In Figure 6a–c, graphs of friction coefficients are given under the wear loads of 10 N, 20 N and 30 N, respectively. It should be stressed that the wear values in all charts start at zero initially, but this is not clearly visible because of the majority of the data transferred to the chart. Having said that, when Figure 6a was examined, it was seen that friction coefficients varied on average between 0.19 and 0.28. The friction coefficients of all materials reached a relative equilibrium state after a very short sliding distance with the start of the wear tests. In other words, it was observed that under 10 N wear load, the first wear-running zone was passed very quickly, and after that the friction coefficient exhibited a stable behavior. Although the friction coefficient of the Ti45Si15Zn20Al20 material reached equilibrium after 50 m of wear distance, it was observed that the friction coefficient values in all other materials reached an equilibrium state after a very sudden increase.

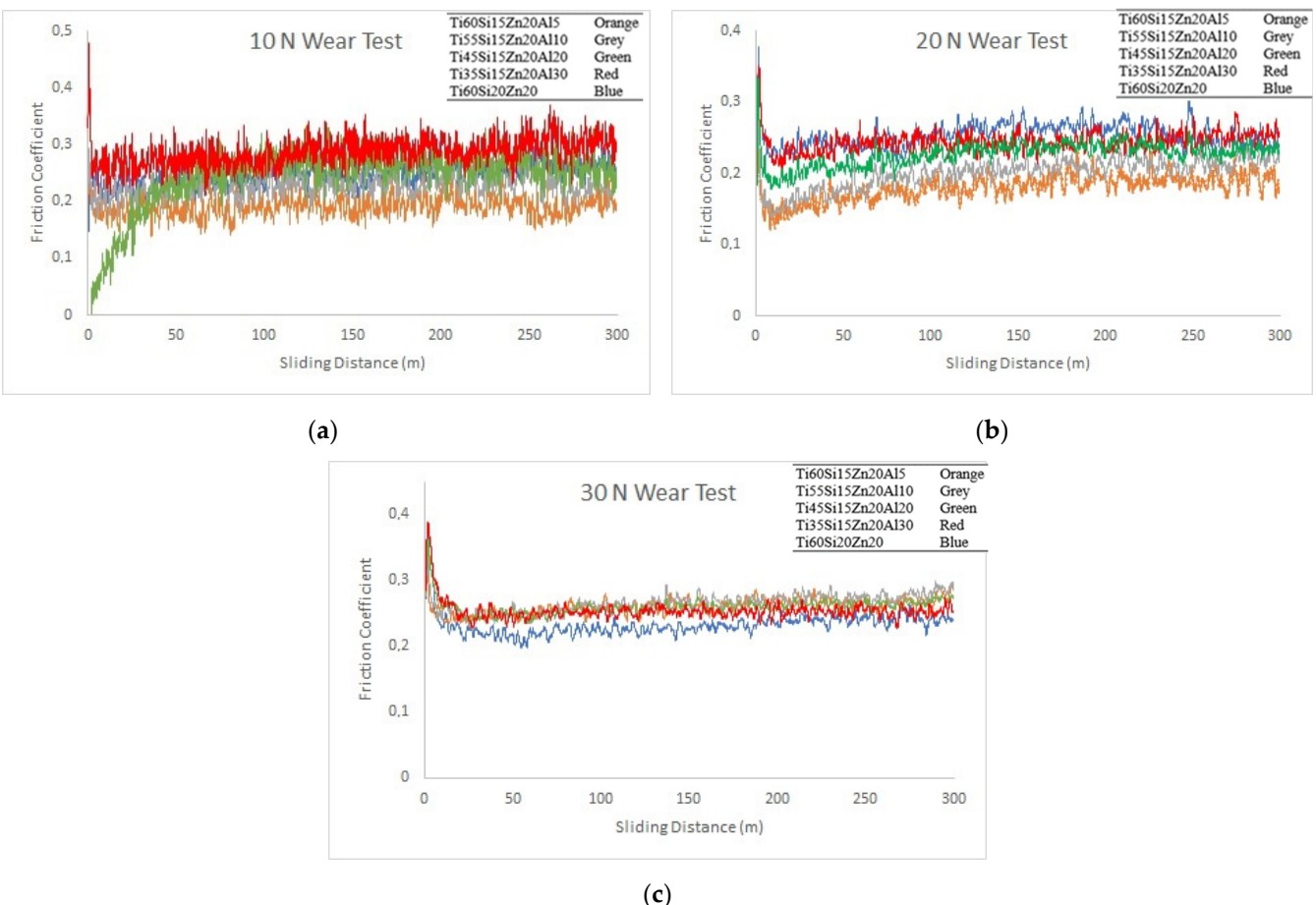

**Figure 6.** (**a**) Friction coefficients of the materials (**a**) under 10 N, (**b**) under 20 N and (**c**) under 30 N wear load, respectively.

When Figure 6b was examined, it was seen that friction coefficients varied on average between 0.14 and 0.45. It was observed that the friction coefficient behavior of the materials under 20 N wear load passed the first wear-running zone very quickly and then showed a stable behavior; the same was true under 10 N wear load. It was found that the friction coefficient values of all materials increased to around 0.35 immediately after starting the wear test and then stabilized on average between 0.15 and 0.25, depending on the material. Furthermore, it was observed that the friction coefficient values under 20 N wear load generally showed similar characteristics with 10 N wear load.

When Figure 6c was examined, it was seen that the friction coefficients varied on average between 0.23 and 0.38 values. It was observed that the friction coefficient values of all materials increased to around 0.35–0.38 right after starting the test and then stabilized

between 0.23 and 0.27 on average. Since the same friction coefficient behavior was reported in refs. [31,46], it should be stressed that the friction coefficient characteristics are compatible with the literature. Furthermore, the friction coefficient behavior of the materials under 30 N wear load showed similar characteristics with the result of 10 N and 20 N wear loads. However, when Figure 6a,b graphs were examined, the friction coefficients of the five materials were around 0.2, while in Figure 6b, they were around 0.25 on average. When all figures were evaluated together, it could be stated that there was no serious effect of increasing alumina content on the friction coefficients of the materials in general.

### 3.5. Hardness

The microhardnesses of materials were measured using Qness Q10M with the conditions of HV3 (3000 gr) and 10 s dwelling time. An example of the hardness measurement can be seen in Figure 7a. When hardness results were reviewed in Figure 7b, it could be stated that the hardness value was around 230 HV for the alumina-free material, and then this value was raised considerably to about 660 HV with the addition of alumina. The highest hardness value, which was over 750 HV, was observed for the Ti45Si15Zn20Al20 material. When the amount of alumina in the compound was increased above 20%, a decrease was encountered, which was around 640 HV, evidenced by the hardness value of the Ti35Si15Zn20Al30 material. The biggest standard deviation, evidenced by the error bars in Figure 7b, was observed in the case of the Ti35Si15Zn20Al30 material.

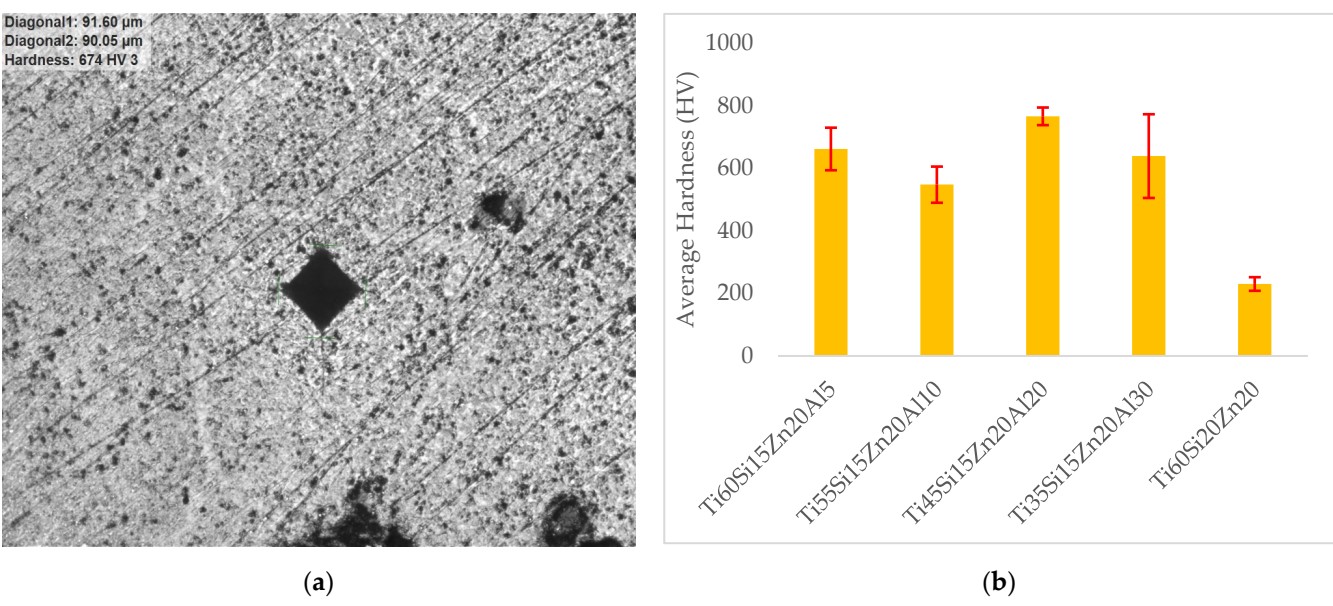

(a)            (b)

**Figure 7.** (**a**) An example of the hardness measurement and (**b**) average hardness of the materials.

### 4. Discussion

In this study, the effects of different weight ratios of alumina addition on the microstructure and wear behavior of the $TiO_2$, $SiO_2$ and ZnO compounds were investigated by using the hot pressing method. The characterization of materials was carried out by using XRD, SEM and EDS. In addition, the materials' morphological and tribological wear behaviors were determined under 10 N, 20 N and 30 N wear loads.

XRD analysis results showed that mostly sharp peaks and phases, signs of crystalline structures, occurred. On the other hand, the presence of broad peaks, a sign of a small crystallite structure, indicates the presence of amorphous structures [38,47]. When the XRD results in Figure 2 were analyzed, it was found that the most severe peaks occurred between 2θ~26.44° and 26.82°. Moreover, it was observed that no significant new peaks were formed depending on the increase in alumina content, and all hot-pressed materials showed similar peaks in XRD analysis.

When the XRD of the materials was examined in terms of the qualitative analysis result of compounds, it was found from Table 2 that rutile ($TiO_2$), spinel ($TiZn_2O_4$), quartz ($SiO_2$), sillimanite ($Al_2SiO_5$), kyanite ($Al_2SiO_5$), mullite ($Al_6Si_2O_{13}$), zincite (ZnO), cristobalite ($SiO_2$) and gahnite ($Al_2ZnO_4$) phases were formed. A few papers containing some of the related phases are refs. [39–41]. In addition, the elements in the general EDS analysis results for each material and the crystalline structure phase formed in XRD analysis were compatible with each other.

When the SEM images were analyzed, it was observed that two phases with complex grain boundaries, manifested by two different inhomogeneous colors, were formed. Apart from the matrix structure, it was determined that a heterogeneous secondary phase structure was constructed at varying heights. While the overall matrix structure was in black, a secondary phase structure (clusters) was not only heterogeneously distributed in the matrix but also differed in size significantly. In addition, it was observed that regions with different heights were formed in both phase structures. The absence of clear grain boundary structures in the SEM images and the formation of multi-layered structures were thought to be because of melting stemming from liquid phase sintering. This phenomenon was also encountered in another study using the same production method for investigating the 1010 steel-based materials containing SiC, MgO and $H_3BO_3$ with varying amounts of $B_4C$ [42].

When the SEM images in Figure 3a,c were examined, it was seen that a heterogeneous secondary phase structure was formed at different heights in a scattered manner apart from the matrix structure. It was observed that the overall matrix structure was black, while secondary phase structures (clusters) were not only heterogeneously distributed in the matrix but distinctively different in size. Contrary to Figure 3a, it was seen that the white regions were not distributed in a manner of significant areas but rather stacked and aggregated. Moreover, it was observed that these clusters were highly reduced among the surface and heterogeneously distributed.

When the SEM images in Figure 3e was examined, it could be emphasized that a heterogeneous biphasic structure was observed from the secondary phase and matrix phase regions in Figure 3e, as in other SEM images. Moreover, it was observed that almost half of the structure was formed from the secondary phase and matrix phase, and the boundaries between these regions were irregular. White clusters were not stacked and aggregated compared with the Ti55Si15Zn20Al10 material. In addition, the secondary phase regions were located at different heights as in the Ti55Si15Zn20Al10 material (Figure 3c). In contrast, the matrix phase regions were formed in a more layered and wavier structure for the Ti45Si15Zn20Al20 material (Figure 3e).

When the SEM image in Figure 3g was analyzed, it was observed that two different phases were formed evidently from the different colored regions. It could be stated that the secondary and matrix phase regions constituted approximately half of the whole structure and showed a heterogeneous distribution. It was observed that a layered structure with no definite grain structure was formed, and complex boundaries were formed between these phases.

When the SEM images in Figure 3a,c,e,g were examined together, it was observed that as the ratio of alumina in the composition increased, the white and black phases transitioned from the dispersed state to a clustered state for each phase separately. This transition was clearly obvious when the SEM images of the Ti45Si15Zn20Al20 material and Ti60Si20Zn20 material were compared. Moreover, when the alumina ratio was increased to 10%, it was determined that the clustered secondary phase regions, which were scattered at different heights and caused a heterogeneous structure, were reduced. It was observed that large layered areas in the secondary phase formations were present in the SEM images of the Ti60Si15Zn20Al5 and Ti45Si15Zn20Al20 materials in contrast to the Ti55Si15Zn20Al10 material. The occurrence of imprecise grain boundaries because of liquid-phase sintering was found in all SEM images [42].

The overall EDS analysis results of materials were compatible with the predetermined weight percentages of the starting elements in the beginning phase. The general EDS analysis results taken from the SEM images showed that mostly Ti and O elements were dominant among chosen regions seen from Figure 3b,d,f,h. Moreover, the EDS analysis results taken from the white and black areas in Figure 3e showed that these regions were dominant in Ti and O elements and Ti and Zn elements, respectively. These results confirmed the argument of two different phase structures occurring in the SEM images.

When the result of the pin-on-disc wear test in Figure 4a was examined, it was determined that similar wear tracks were formed under different wear loads for the Ti60Si15Zn20Al5 material. It was observed that as the load increased, the width of the wear tracks increased accordingly, and ruptures occurred on the wear trajectory. Similar observations were stated in ref. [46]. In addition, it was also determined that under some wear loads, plowed formations (points a and b in Figure 4a) in place of the ruptured parts occurred. It should be stressed that these ruptures could be ascribed to the weak bonding and porosities between the hot-pressed structures [31].

It was observed that the Ti55Si15Zn20Al10 material showed similar wear tracks under all wear loads (Figure 4b). It was found that as the wear load increased, the depth and width of the wear tracks and plowed structures on the worn surface occurring because of ruptures increased accordingly, as in the Ti60Si15Zn20Al5 material. The ruptured parts broke off in the wear trajectory resulted in a grooved structure. The grooves on the worn surfaces in the wear test indicate the abrasive and adhesive wear mechanism [45,48]. It was further determined that as the wear load increased, the wear track became more pronounced. This phenomenon is a conclusion of the possible oxide compounds formed on the worn surfaces causing plastic deformation during wear and resulting in the removal of materials from the worn surfaces due to adhesion and cohesive failure [45,49–51].

It was observed in Figure 4c that Ti45Si15Zn20Al20 material showed similar wear marks under all loads, just like the wear behavior in Figure 4a,b. It was determined that the wear depth of this material was relatively lower at a low wear load, although there were pieces that broke off in the wear trajectory as in other materials at bigger loads. Moreover, it was observed that the wear tracks were formed in a grooved structure, resulting in plastic deformation zones as the broken parts remaining in the wear trajectory zone [52].

When the SEM image of the Ti60Si20Zn20 material in Figure 4d was examined, it was observed that similar wear tracks were formed as in the previous materials, and the depth and width of the wear tracks were increased as the load increased. It was determined in Figure 4d that the plowed formations due to the broken parts in the wear zone were more than in the previous materials under all loads. In addition, it was further observed that the wear track width of Ti60Si20Zn20 material was generally larger than the previous materials discussed. It is thought that the reason for this phenomenon is the decrease in the matrix phase regions according to the increase in the alumina content. Post-wear weight losses given in Figure 5 and SEM images are support this hypothesis and are compatible with each other. Moreover, plastic deformation zones were found in wear tests for all materials containing alumina, especially at the wear load of 30 N. This behavior is predominantly attributed to the occurrence of the removal of sheet/plate-like parts from the surface of the worn material, causing a large plastic flow, as predicted in ref. [44].

In the literature, when the studies investigating the wear behavior of different materials containing alumina at varying ratios were examined, it was reported that the wear track and depth were increased because of the increase in wear load [26,46,53–56]. Therefore, the argument that the wear track and depth increase depending on the wear load for the same material, which is seen in the wear behavior of all materials, is in line with the literature.

When the weight losses of each material under all wear loads in Figure 5 were examined, it was determined that the highest weight losses occurred in the Ti55Si15Zn20Al10 material. The probable reason for this occurrence was considered to stem from the lesser matrix phase regions on the surface compared with other materials with large layered areas of secondary phases. Therefore, the highest weight loss in the Ti55Si15Zn20Al10

material was attributed to the large fragments being broken off during the wear test from the secondary, i.e., white, region phases where the secondary phase structure regions were more clustered on the surface regarding other materials. It was further observed that the weight losses decreased as the $Al_2O_3$ ratios in $TiO_2$, $SiO_2$ and ZnO compounds increased up to 20 and 30 wt%. The wear test results showed that the weight loss was the lowest in Ti35Si15Zn20Al30 material. The most probable reason for this phenomenon is the increased contact area occurrence, which leads to a bigger load carrying capacity, with the inclusion of alumina compared with other materials. Having a bigger load carrying capacity results in increased wear resistance accordingly. Identical interpretations were reported in ref. [52].

When Figure 6a–c were examined, although friction coefficients in all materials generally fluctuated, friction coefficient values showed a stable behavior after the first 50 m of sliding distance. In other words, during the wear test of the materials under 10 N, 20 N and 30 N wear loads, the friction coefficients reached 0.35–0.45 values with a very sudden increase from zero, and the tests were concluded as if there were zig-zag movements around a line that could be considered as stable. This behavior was generally observed in all materials. The same friction coefficient behavior was reported in ref. [46]. In addition, it was determined, in general, that with the increase of alumina content, the friction coefficients were not proportional because of a slight increase at 30 N wear load and a slight decrease at 20 N wear load regarding 10 N wear load. Therefore, a direct relationship between the friction coefficient and increasing alumina content in compounds could not be identified. Similar results were reported in ref. [46], emphasizing that the alumina concentration had a negligible effect on the friction coefficients.

When Figure 7b was examined, it could be observed that there was a substantial increase in the microhardness value, generally depending on the increase in alumina content. The lowest hardness value was encountered in the case of Ti60Si20Zn20 material, and the highest hardness value was observed in the Ti45Si15Zn20Al20 material. When the alumina content was raised over 20% in the compounds, it was seen that the hardness value decreased, as evidenced by the hardness value of the Ti35Si15Zn20Al30 material. The highest standard deviation of hardness value was obtained for the Ti35Si15Zn20Al30 material. The results showed that the material with the highest microhardness value generally showed the minimum weight loss in the wear test. Therefore, it can be stressed that as the $Al_2O_3$ increased over 10 wt%, the weight losses were generally decreased. Thus, it can be stated that the microhardness results and weight losses of the wear test are generally in line with each other [46,52,57].

Gugulothu et al. [57] investigated the combined effect of varying amounts of alumina with constant wt% $ZrO_2$ on the wear behavior of Al5052 alloy. The Taguchi method was used for parametric optimization of wear behavior, and it was found that the leading parameter influencing the wear behavior of the materials was the load parameter. Bharath et al. [52] examined the effect of alumina amounts on the wear behavior of Al2014 alloy and stated that alumina-reinforced Al2014 alloy composites showed lower friction coefficients and better wear resistance compared with plain Al2014 alloy. Al-Qutup et al. stated in a study that wear rate increased thanks to higher concentration amounts of alumina in the matrix structure [46]. This situation was attributed to the increase in the abrasive wear mechanism in proportion to the increase in the alumina content in the matrix structure. Therefore, in light of the citations, it can be stated that the results of this study are in agreement with the literature.

## 5. Conclusions

In this study, varying amounts of alumina effect such as 0, 5, 10, 20 and 30 wt% in $TiO_2$, $SiO_2$ and ZnO compounds were considered to investigate the wear characteristics of hot-pressed materials under 10 N, 20 N and 30 N wear loads. The conclusions drawn from this study could be summarized as follows:

1. According to XRD results, it was observed that a crystalline structure was formed in all materials and no new peaks were formed depending on the increase in alumina content;

2. Rutile, spinel, quartz, sillimanite, kyanite, mullite, zincite, cristobalite and gahnite phases were determined in XRD results. The resulting phases were compatible with the starting compounds;

3. It was observed that two complex phases with unclear grain boundaries were formed in the SEM results. The reduction of crystallinity in the XRD results could not be identified in the SEM images because of liquid phase sintering;

4. The elements in the general EDS analysis and the crystalline phases formed in the XRD analysis were consistent with each other. Additionally, overall EDS analysis results were found to be compatible with the weight percentages determined in the beginning phase;

5. It was observed that the wear track and depth were increased as the wear load increased for the same material. Besides, in general, the $Al_2O_3$ contents increasing with wear track and depth were found to be inversely proportional under the same wear load;

6. The highest weight loss under different wear loads was obtained in the Ti55Si15Zn20Al10 material. It was determined that as the $Al_2O_3$ wt% increased in the compositions, the weight losses decreased, and the microhardness values increased generally;

7. As a result of weight measurements taken before and after wear tests, the highest weight loss was observed at 30 N in the material with 10 percent alumina content. When the alumina content was increased to 20 percent and 30 percent, it was observed that the weight losses decreased because of possible oxide compounds formed on the worn surfaces, causing plastic deformation;

8. When the hardness measurement results were examined, it was observed that the highest value occurred in the material containing 20 percent alumina, while the lowest value occurred in the material without alumina. When the weight losses of wear tests and microhardness results were evaluated together, it was concluded that the optimum alumina content was 20 percent.

**Funding:** This research received no external funding.

**Data Availability Statement:** The related data as discussed in this article can be requested from the corresponding author.

**Conflicts of Interest:** The authors declare no conflict of interest.

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
