# Peer review of "Influence of Varying Amounts of Alumina (Al2O3) on the Wear Behavior of ZnO, SiO2 and TiO2 Compounds"

_processes, doi:10.3390/pr11041073_

Round 1
Reviewer 1 Report
This is a very well-written paper in terms of structure, results and technical presentation format. However, the English of the paper should be edited.
Author Response
Dear Mr(s). Reviewer
First of all, I would like to thank the reviewers for their thoughts on the study. I would also like to thank them for their valuable criticisms and questions to increase the academic depth and originality of this study. All the changes I have made on the study have been tracked and saved thanks to the relevant feature of the "Word" program. The reviewers can observe these changes themselves if they wish. Below, you can see my responses to each reviewer separately on the issues requested to be corrected or improved.
Best regards
Reviewer 1:
This is a very well-written paper in terms of structure, results and technical presentation format. However, the English of the paper should be edited.
Answer to Reviewer 1:
I would like to thank the referee for his worthful thoughts on the study, and I would like to emphasize that I edited the study both by purchasing editing services from a native English editor and by editing it with a grammar check program.
Reviewer 2 Report
Line 169: X-ray diffraction analysis is the determination of different phases in multi-phase materials - it is methodic that allows to determinate phases in multi-phase materials, but not only, sentence is not correct.
Line 179: not peek but peak
Line 194-195: The sharp peaks in all XRD patterns indicate that the compounds at different 2θ positions are in crystalline structure - meaning of the sentence is not clear
Figure 3,4 5 and 6 could be combined in one figure, for all figures b) image should be rescale without empty part of the spectrum
Figure 7 and 8 should be combined
Author Response
Dear Mr(s). Reviewer
First of all, I would like to thank the reviewers for their thoughts on the study. I would also like to thank them for their valuable criticisms and questions to increase the academic depth and originality of this study. All the changes I have made on the study have been tracked and saved thanks to the relevant feature of the "Word" program. The reviewers can observe these changes themselves if they wish. Below, you can see my responses to each reviewer separately on the issues requested to be corrected or improved.
Best regards
Reviewer 2:
- Line 169: X-ray diffraction analysis is the determination of different phases in multi-phase materials - it is methodic that allows to determinate phases in multi-phase materials, but not only, sentence is not correct.
- Line 179: not peek but peak
- Line 194-195: The sharp peaks in all XRD patterns indicate that the compounds at different 2θ positions are in crystalline structure - meaning of the sentence is not clear
- Figure 3, 4 5 and 6 could be combined in one figure, for all figures b) image should be rescale without empty part of the spectrum
- Figure 7 and 8 should be combined
Answer to Reviewer 2:
- The line corrected as “X-ray diffraction analysis is the quantitative analysis of multi-phase materials [37]” and the following academic publication was cited to support the statement.
Popović, S. Quantitative Phase Analysis by X-ray Diffraction—Doping Methods and Applications. Crystals 2020 10(1), 27. https://doi.org/10.3390/cryst10010027
- I would like to apologize for this simple error, and all words in which this overlooked error occurred have been replaced with the correct wording as requested by reviewer.
- The sentence is changed as “The sharp peaks in all XRD patterns indicate the crystalline structure positions of different phases in the materials”.
- Figure 3, 4 5 and 6 were combined in one figure and “b)” images were reorganized in order to eliminate the white space areas in the spectrum contained in the “b)” figures.
- Figure 7 and 8 were combined and all figure numbers in the study have been rearranged according to these changes.
Reviewer 3 Report
Review of “ Influence of Varying Amount of Alumina (Al2O3) on the Wear Behavior of ZnO, SiO2 and TiO2 Compounds" by Ali Ihsan Kaya
The theme of this paper is good. However, the conclusions can be written in a better way. The text needs careful revision.
Author Response
Dear Mr(s). Reviewer
First of all, I would like to thank the reviewers for their thoughts on the study. I would also like to thank them for their valuable criticisms and questions to increase the academic depth and originality of this study. All the changes I have made on the study have been tracked and saved thanks to the relevant feature of the "Word" program. The reviewers can observe these changes themselves if they wish. Below, you can see my responses to each reviewer separately on the issues requested to be corrected or improved.
Best regards
Reviewer 3:
The theme of this paper is good. However, the conclusions can be written in a better way. The text needs careful revision.
Answer to Reviewer 3:
I would like to thank the referee for his worthful thoughts on the study, and I would like to emphasize that I edited the study both by purchasing editing services from a native English editor and by editing it with a grammar check program. Moreover, in the conclusion;
- Item 7 in the concluding section is amended as “As a result of weight measurements taken before and after wear tests, the highest weight loss was observed at 30 N in the material with 10 percent alumina content. When the alumina content was increased to 20 percent and 30 percent, it was observed that the weight losses decreased due to possible oxide compounds formed on the worn surfaces causing plastic deformation” for discussion in terms of weight loss. Besides,
- Item 8 was added “When the hardness measurement results were examined, it was observed that the highest value was observed in the material containing 20 percent alumina, while the lowest value was observed in the material without alumina. When the weight losses of wear tests and microhardness results were evaluated together, it was obtained that the optimum alumina content was 20 percent.” For discussion of optimum content of alumina.
changes were made.
Reviewer 4 Report
The manuscript entitled "Influence of Varying Amount of Alumina (Al2O3) on the Wear 2 Behavior of ZnO, SiO2 and TiO2 Compounds" has significant importance and contributes to the field. However, the manuscript could be better arranged, and some expressions could be more transparent. The authors should revise the manuscript addressing the following comments.
Please highlight the need for research in the abstract with a sentence showing the background and the current research gap.
The author has given an excellent literature review based on the background of the study in the introduction part. However, there is no connection or a particular link that connects each paragraph in the introduction. Please logically connect each paragraph in the Introduction section to make the section more attractive.
Please highlight the novelty of this work in the introduction section.
In Table 1, the author represents the different weight percentage variations of TiO2 and Al2O3. In addition, there are variations in ZnO also. The title of this manuscript implies the "Influence of Varying Amount of Alumina (Al2O3) on the Wear Behavior of ZnO, SiO2 and TiO2 Compounds". The authors mentioned the different weight percentages of the compounds in this manuscript. How can the effect of varying wt% of TiO2 be determined from the formulation used in this manuscript? Please add clarification and supporting points to the modified manuscript.
Please add the reference of standards followed for the mechanical testing.
I suggest adding discussions together with the results. So the readers easily compare the results from the figures and their following discussions.
From Figure 9, "It was further observed that the weight losses decreased as the Al2O3 ratio in TiO2, SiO2 and ZnO compounds increased up to 20% and 30% wt. How can Al2O3 control the weight loss of the compounds?
"It was determined that as the Al2O3 wt% increase in the 567 compositions, the weight losses were decreased and the micro-hardness values were increased generally". How much weight of Al2O3, and what should be the minimum? Please discuss the optimum weight concentration of Al2O3 content to be used for better properties.
Author Response
Dear Mr(s). Reviewer
First of all, I would like to thank the reviewers for their thoughts on the study. I would also like to thank them for their valuable criticisms and questions to increase the academic depth and originality of this study. All the changes I have made on the study have been tracked and saved thanks to the relevant feature of the "Word" program. The reviewers can observe these changes themselves if they wish. Below, you can see my responses to each reviewer separately on the issues requested to be corrected or improved.
Best regards
Reviewer 4:
The manuscript entitled "Influence of Varying Amount of Alumina (Al2O3) on the Wear 2 Behavior of ZnO, SiO2 and TiO2 Compounds" has significant importance and contributes to the field. However, the manuscript could be better arranged, and some expressions could be more transparent. The authors should revise the manuscript addressing the following comments.
- Please highlight the need for research in the abstract with a sentence showing the background and the current research gap.
- The author has given an excellent literature review based on the background of the study in the introduction part. However, there is no connection or a particular link that connects each paragraph in the introduction. Please logically connect each paragraph in the Introduction section to make the section more attractive.
- Please highlight the novelty of this work in the introduction section.
- In Table 1, the author represents the different weight percentage variations of TiO2 and Al2O3. In addition, there are variations in ZnO also. The title of this manuscript implies the "Influence of Varying Amount of Alumina (Al2O3) on the Wear Behavior of ZnO, SiO2 and TiO2 Compounds". The authors mentioned the different weight percentages of the compounds in this manuscript. How can the effect of varying wt% of TiO2 be determined from the formulation used in this manuscript? Please add clarification and supporting points to the modified manuscript.
- Please add the reference of standards followed for the mechanical testing.
- I suggest adding discussions together with the results. So the readers easily compare the results from the figures and their following discussions.
- From Figure 9, "It was further observed that the weight losses decreased as the Al2O3 ratio in TiO2, SiO2 and ZnO compounds increased up to 20% and 30% wt. How can Al2O3 control the weight loss of the compounds?
- "It was determined that as the Al2O3 wt% increase in the 567 compositions, the weight losses were decreased and the micro-hardness values were increased generally". How much weight of Al2O3, and what should be the minimum? Please discuss the optimum weight concentration of Al2O3 content to be used for better properties.
Answer to Reviewer 4:
- In the abstract a sentence “This study aimed to exploit the superior properties of TiO2, ZnO, SiO2 and Al2O3 inorganic materials to combine them under pressure and investigate their mechanical properties.” was added for showing the background and the current research gap starting from line 8.
- In order to connect each paragraph in the introduction:
- “Particle size is one of the important factors in powder metallurgy. For example,…” sentence was to at the beginning of second paragraph of introduction, line 31.
- “In biomedical applications, Ti and its alloys are commonly utilized as medical implants for a variety of purposes whereas ZnO material is used in pharmaceutical and cosmetic uses.” sentence was to at the beginning of third paragraph of introduction, line 50.
- “Due to their biocompatibility and the beneficial biological effects, silica-based materials are another biomaterial widely used in biomedical applications like TiO2 and ZnO.” sentence was to at the beginning of fourth paragraph of introduction, line 68.
- “Unlike TiO2, ZnO, and SiO2, alumina was the first biomaterial to be used in a clin-ical applications because of its excellent biocompatibility, hardness, strength to with-stand fatigue, and corrosion resistance.” sentence was to at the beginning of fourth paragraph of introduction, line 83.
- The sentence “When the studies on biomaterials are examined, it has been observed that TiO2, ZnO, SiO2, and alumina make considerable contributions to this field in terms of both chemical structure and mechanical properties. However, to the best of author’s knowledge, no academic study has been observed in which the properties of these four biomaterials are investigated by combining them together.” was added to the beginning of last paragraph at the end of Introduction part starting from line 119.
- “As can be seen in Table 1, new materials were produced by considering the weight percentages of different biomaterials in this study. Therefore, depending on this calcu-lation, one of the existing materials must be reduced by weight in order to look at the effect of another doped material. Since ZnO material additive improves wear re-sistance and silica improves mechanical properties, the ratios of these materials were tried to be kept as constant as possible. For this reason, one material, in this case TiO2, was considered as a matrix structure and its ratios were changed as per alumina addi-tion. In addition, while producing biomaterials, care was taken to ensure that the weight percentages of the additives were lower by weight than the TiO2 determined as the matrix.” Sentence was added to line 150 for clarification of weight percentages changes of alumina and TiO2.
- “operating according to ASTM G99-17” statement was added to related wear test under the subheading of Pin-on-Disc Wear Testing.
- In order to be faithful to the main headings in the template file of the processes journal, the results and discussions sections have been organised separately in this way. But the following paragraphs have been moved to the relevant places in the discussion section as suggested by reviewer.
- It was seen that a heterogeneous secondary phase structure was formed at different heights in a scattered manner apart from the matrix structure. It was observed that the overall matrix structure was black, while secondary phase structures (clusters) were not only heterogeneously distributed in the matrix but distinctively different in size. Contrary to Figure 3a, it was seen that the white regions were not distributed in a manner of significant areas but rather stacked and aggregated. Moreover, it was observed that these clusters were highly reduced among the surface and heterogeneously distributed.
- It could be emphasized that a heterogeneous biphasic structure was observed evidently from the secondary phase and matrix phase regions in Figure 3e, as in other SEM im-ages. Moreover, it was observed that almost half of the structure was formed from the secondary phase and matrix phase, and the boundaries between these regions were ir-regular. White clusters were not stacked and aggregated on contrary to the Ti55Si15Zn20Al10 material. In addition, the secondary phase regions were located at different heights as in the Ti55Si15Zn20Al10 material (Figure 3c). In contrast, the ma-trix phase regions were formed in a more layered and wavier structure for the Ti45Si15Zn20Al20 material (Figure 3e).
- When the SEM image in Figure 3g was analyzed, it was observed that two differ-ent phases were formed evidently from the different colored regions. It could be stated that the secondary and matrix phase regions constitute approximately half of the whole structure and show a heterogeneous distribution. It was observed that a layered structure with no definite grain structure was formed, and complex boundaries were formed between these phases.
- As alumina was added to the TiO2, ZnO, and Silica biomaterials, it was observed in the SEM images that the secondary phases increased in the surface areas of produced materials. This can be seen by examining Figures 3g, 3a, 3c, and 3e, respectively. Besides, the contribution of alumina to the wear resistance of the current biomaterials can be seen from the increase in the microhardness of the biomaterials produced. As the alumina increased in the compounds, microhardness increased accordingly, evidently from Figure 6. Therefore, these phenomena show the effect of alumina on weight loss.
- Item 7 in the concluding section is amended as “As a result of weight measurements taken before and after wear tests, the highest weight loss was observed at 30 N in the material with 10 percent alumina content. When the alumina content was increased to 20 percent and 30 percent, it was observed that the weight losses decreased due to possible oxide compounds formed on the worn surfaces causing plastic deformation” for discussion in terms of weight loss. Besides item 8 was added “When the hardness measurement results were examined, it was observed that the highest value was observed in the material containing 20 percent alumina, while the lowest value was observed in the material without alumina. When the weight losses of wear tests and microhardness results were evaluated together, it was obtained that the optimum alumina content was 20 percent.” For discussion of optimum content of alumina.
Round 2
Reviewer 4 Report
The author made relevant corrections. I recommend accepting the manuscript for publication.